# The Respective Effects of Vapor Pressure Deficit and Soil Moisture on Ecosystem Productivity in Southwest China

**Xupeng Sun** [1], **Yao Xiao** [1], **Jinghan Wang** [1], **Miaohang Zhou** [1], **Zengjing Song** [1,2], **Mingguo Ma** [1] and **Xujun Han** [1,*]

1    Chongqing Engineering Research Center for Remote Sensing Big Data Application, Chongqing Jinfo Mountain National Field Scientific Observation and Research Station for Karst Ecosystem, School of Geographical Sciences, Southwest University, Chongqing 400715, China; giser2020@email.swu.edu.cn (X.S.); xy10086@email.swu.edu.cn (Y.X.); jankinwong@email.swu.edu.cn (J.W.); zmh18981434893@email.swu.edu.cn (M.Z.); z.song@utwente.nl (Z.S.); mmg@swu.edu.cn (M.M.)
2    Faculty of Geo-Information Science and Earth Observation (ITC), University of Twente, 7522 NB Enschede, The Netherlands
*    Correspondence: hanxujun@swu.edu.cn; Tel.: +86-023-68367339

**Abstract:** This study aims to examine the individual and combined effects of soil moisture (SM) and vapor pressure deficit (VPD) on ecosystem productivity in Southwest China. Utilizing the community land model (CLM) to simulate the regional soil moisture and vapor pressure deficit, we analyzed their impacts on ecosystem productivity through a data binning approach and employed sun-induced chlorophyll fluorescence yield (SIFyield) as a productivity indicator. Our findings highlight a significant coupling effect between SM and VPD, which diminishes with finer temporal data resolution. The data binning analysis indicates that VPD has a predominant influence on SIFyield across 70% of the study area, whereas SM is more influential in the remaining 30%. Notably, the correlation between SIFyield and SM, modulated by VPD, is stronger in forest and shrubland ecosystems, whereas in grasslands, the influence pattern is reversed, with VPD having a more significant impact. The study concludes that in Southwest China, ecosystem productivity is more significantly affected by VPD than by SM.

**Keywords:** soil moisture; vapor pressure difference; SIF; drought

## 1. Introduction

The productive capacity of an ecosystem is a fundamental metric for examining the carbon balance and water use efficiency of a regional ecosystem [1], and it is an essential parameter in the carbon–water coupling of such ecosystems [2]. Over the past few years, the occurrence of droughts has become increasingly common on a global scale, resulting in devastating damage to regional ecosystems [3–5]. The cumulative impact of extreme drought can severely compromise the production capacity of ecosystems and, consequently, have a significant effect on the carbon sink of those ecosystems [6,7]. Indeed, ecosystem production is a key indicator of how ecosystems are responding to changing environmental conditions in the wider context of climate change [8,9]. As a result, it has become an essential parameter in the assessment of the health and resilience of regional ecosystems. In the context of climate change, quantitative research on factors affecting regional ecosystem productivity will help us to deepen our understanding of the terrestrial–atmospheric carbon and water cycles.

Soil moisture and vapor pressure deficit, respectively, are deemed to be one of the important drivers of regional ecosystem production [10]. From a vegetation physiology viewpoint, low soil moisture and high vapor pressure deficit both cause vegetation to suffer from drought stress [11], which affects the normal physiological activities of vegetation and, hence, causes the death of vegetation and impairs the productivity of the ecosystem [12].

There has been significant debate regarding the respective contributions of soil moisture and atmospheric water demand in studies investigating the response of vegetation to drought [10]. VPD causes plant stomata to close. This controls physiological processes such as transpiration and photosynthesis [13]. Excessive VPD induces vegetation stomatal closure, thus limiting photosynthesis in vegetation, which in turn leads to a decrease in ecosystem productivity [14]. Soil moisture is the direct source of water for vegetation. Low soil moisture leads to agricultural drought and even vegetation death [15]. A comprehensive understanding of the roles played by soil moisture and vapor pressure deficits in ecosystem production is essential. The results are necessary for accurately assessing the impacts of drought on regional ecosystems.

Examining the respective impacts of soil moisture and vapor pressure deficit on ecosystem production at a regional scale can be challenging. Soil moisture and vapor pressure deficit are intricately linked [16], necessitating distinct methodologies to individually assess their impacts on ecosystem productivity. Furthermore, factors such as temperature, solar radiation, and rainfall play critical roles in regional ecosystem productivity [17]. It is essential to isolate the impacts of these elements when examining the specific effects of soil moisture and vapor pressure deficit on vegetation productivity. However, such research is vital for understanding ecosystem response mechanisms to drought and promoting environmental protection. Liu conducted a global-scale investigation of the effects of low SM and high VPD on ecosystem production using a data splitting box approach [10]. It was discovered that SM plays a dominant role in most terrestrial vegetation ecosystem production subjected to drought stress compared to VPD. Lu contends that Liu's study does not account for the influence of photosynthetically active radiation on VPD and SM, leading to inaccuracies in the findings. They argue that the effect of VPD on ecosystem productivity is at least as significant as that of SM [18]. Existing research highlights the importance of examining the impact of soil moisture and vapor pressure deficit on ecosystem productivity at a regional scale. This investigation is crucial for enhancing our understanding of ecosystem responses to meteorological factors.

Southwest China is the major karst landscape distribution area in China, with extremely fragile ecosystems that are strongly vulnerable to extreme hazards [19]. In recent years, there has been a noticeable increase in the frequency and intensity of drought events in Southwest China [20], causing significant negative impacts on regional ecosystem production as well as the sustainable development of human society [21]. Research on the impact of different drivers on ecosystem production and carbon sinks in Southwest China has been the focus of academic interest [22–24]. Chen et al. investigated the effects of drought on vegetation productivity in Southwest China utilizing sun-induced chlorophyll fluorescence (SIF) and soil moisture. The results of the research indicate that large-scale drought significantly influences regional ecosystem productivity [25]. Chen et al. analyzed the impact of VPD on regional primary productivity in three typical ecosystem study areas in China and revealed that the decline in GPP in Southwest China was closely related to SM and VPD, and that more than 50% of the change in GPP was attributable to the combination of SM and VPD [26]. The above results demonstrate the importance of SM and VPD in the study of the productive capacity of regional ecosystems in response to climate elements. Despite the numerous studies on SM's and VPD's effects on ecosystem productivity [18,27], fewer studies have clarified the respective influences of SM and VPD on ecosystem production. Therefore, we need to clarify the impact of SM and VPD on ecosystem productivity in Southwest China. Determining the main factors that dominate vegetation productivity in SM and VPD will help us to deepen our understanding of the carbon water cycle in the ecosystem.

This study aims to examine the respective impact of soil moisture and vapor pressure deficit on ecosystem production in Southwest China. To achieve this goal, we first simulated soil moisture and other variables in Southwest China using the CLM4.5 model in conjunction with GLDAS atmosphere-driven data and verified the accuracy of the simulation results. Then, we decoupled SM and VPD using the data split-box approach and

quantitatively assessed the extent of influence of each of SM and VPD on the productivity of vegetation in the southwest. Finally, we investigated the effect of changes in SM and VPD on the correlation between each other and ecosystem productivity, as well as the variability of this result under different vegetation types. The results of this study are expected to provide scientific suggestions and a theoretical foundation for addressing future droughts in Southwest China.

## 2. Materials and Methods

In this study, the study area's soil moisture and other variables were simulated using the land surface process model CLM4.5 from 2005 to 2020. The CLM4.5 model is an internationally recognized and highly advanced model for studying land surface processes. It is the land surface module of the Common Earth System Model [28,29]. The CLM4.5 model realizes the heterogeneity of land surface space using a nesting grid. The grid contains a variety of land individuals, snow, soil columns, and vegetation functional types [30]. The CLM4.5 model stands out due to its nested grid capability, which enables the simulation of intricate surface features. Moreover, it offers a detailed and complex depiction of processes such as soil moisture dynamics, plant growth and mortality, and the exchange of water and energy between vegetation and the atmosphere [31]. CLM4.5 encompasses extensive simulations of phenomena like root water uptake, soil evaporation, plant transpiration, and permafrost processes [32]. As a result, it potentially delivers more accurate simulations of soil moisture dynamics under specific conditions compared to other models. In this study, we use the CLM4.5 model to simulate soil moisture and evapotranspiration data in Southwest China.

Simultaneous changes in photosynthetically active radiation (PAR) and fraction of photosynthetically active radiation (fPAR) with VPD and SM interfere with the analysis of VPD and SM effects [33,34]. For example, when VPD rises, the accompanying rise in PAR boosts ecosystem production. That would offset the decline in ecosystem production caused by a rise in VPD [18]. With regard to SM, both FPAR and SM reductions equally cause a reduction in SIF. To better separate the simultaneous effects of PAR and FPAR on ecosystem production, the experiment was conducted with Yuan's method of applying SIFyield instead of SIF [27]. SIFyield was calculated as follows:

$$\text{SIF}_{\text{yield}} = \frac{\text{SIF}}{\text{PAR} \times \text{fPAR}} \tag{1}$$

SM and VPD are strongly coupled, and the experiment requires a reduction in the coupling of the two elements to better investigate the influence of each on regional ecosystem production [10]. This experiment employs a data-binning approach [35,36] to reduce the strong coupling correlation between SM and VPD, which is performed as follows. The data binning process is outlined as follows: For each pixel, we established the 10th, 20th, and up to the 100th percentile thresholds for both SM and VPD. These thresholds were utilized to categorize the data into 10 distinct bins, corresponding to the percentile ranges of 0–10th, 10th–20th, through to 80–90th, and 90–100th for SM or VPD. This binning procedure maintained the temporal alignment of the datasets. Given that SM and VPD were substantially uncoupled within each specific SM or VPD bin, we were able to isolate the individual impacts of SM and VPD on the SIFyield. To ensure comparability across different spatial areas, the SIFyield time series for each pixel was normalized against the average SIFyield observed at the 90th percentile for that pixel.

The effects of SM and VPD on SIFyield [18] were counted separately on the basis of data bins, where the effect of VPD on SIFyield was calculated as follows:

$$\Delta\text{SIFyield}(\text{VPD}|\text{SM}) = \frac{1}{N}\sum_{k=0}^{n}\text{SIFyield}_{k,m_{k,\max}} - \text{SIFyield}_{k,m_{k,\min}} \tag{2}$$

where N is the number of bins of SM, k is a particular bin number of SM, and $m_{k,max}$ and $m_{k,min}$ are the maximum and minimum VPD bin numbers of the bins for which SM is k. The meaning of Equation (2) is to calculate the difference between the SIFyield corresponding to the largest bins number with VPD and the smallest bins number for all SM bins, which is the change in SIFyield variation due to high VPD.

$$\Delta \text{SIFyield}(\text{SM}|\text{VPD}) = \frac{1}{N}\sum_{l=0}^{n} \text{SIFyield}_{m_{l,min},l} - \text{SIFyield}_{m_{l,max},l} \tag{3}$$

where N is the number of bins of VPD, l is a particular bin number of VPD, and $m_{l,max}$ and $m_{l,min}$ are the maximum and minimum SM bin numbers of the bins for which VPD is l. The meaning of Equation (3) is to calculate the difference between the SIFyield corresponding to the smallest sub-box number of the SM and the largest sub-box number for all VPD sub-boxes, which is the change in SIFyield variation caused by the low SM.

Temperature, precipitation, and solar radiation are important factors affecting the productivity of vegetated ecosystems, but we need to avoid the influence of the above factors on our study to better investigate the respective effects of SM and VPD on the productivity of vegetated ecosystems. Therefore, we performed some data screening before data binning to help us avoid the influence of temperature and other factors. Specifically, we need the following steps:

(1) Consider only pixels with daily average temperature greater than 15 °C, and exclude pixels that do not meet the requirements of SM, VPD, and the corresponding time of SIFyield according to the conditions;

(2) Consider only pixels with daily average VPD greater than 0.5 kPa, and exclude pixels that do not meet the requirements of SM, VPD, and the corresponding time of SIFyield according to the conditions.

The above operation can effectively avoid the influences of temperature as well as solar radiation on vegetation productivity. The high-temperature period in Southwest China is accompanied by high rainfall, and the temperature screening also avoids the effect of rainfall to some extent [10,18].

Solar-induced chlorophyll fluorescence (SIF) has significant potential as an effective indicator for monitoring gross primary productivity (GPP) and evaluating plant photosynthesis [37]. The relationship between SIF and GPP on a global scale stated that GPP had a significant positive correlation with SIF ($p < 0.001$) [38]. In this experiment, the global high-resolution SIF data produced by Chen [39] were selected as an indicator of regional ecosystem production. The data presented were generated using the XGBoost machine learning model, with a temporal resolution of 8 days and a spatial resolution of 0.05°. The SIF data obtained had both a long temporal range and high spatial resolution, making it highly valuable for evaluating long-term terrestrial ecosystem photosynthesis and global carbon water flux.

Additionally, the photosynthetically available radiation (PAR) and fraction of absorbed photosynthetically active radiation (fPAR) products were used, which were obtained from the Global Land Surface Satellite (GLASS) [40]. The production of these two products is based on a look-up table approach which, combined with Moderate Resolution Imaging Spectroradiometer (MODIS) data, generates land surface data products with global coverage. Both the PAR and fPAR data have a temporal resolution of 1 day and a spatial resolution of 0.05°, spanning from 2000 to 2021. These data products are commonly used in studies related to the carbon cycle and carbon driving mechanisms.

The CLM atmospheric drive data were derived from GLDAS-2.1 with a spatial resolution of 0.25° and a temporal resolution of 3 h. The atmospheric drive data consist mainly of data on precipitation, air temperature, barometric radiation, etc. Land Cover data is produced by MODIS with International Geosphere–Biosphere Programme (IGBP) classification standard and a spatial resolution of 500 m [41].

### 3. Study Area

Southwest China mainly includes Yunnan, Sichuan, Guizhou, Guangxi, and Chongqing Provinces (Figure 1) [42]. The entire study area lies between 94°21′E–112°04′E and 20°54′E–34°19′N at an altitude of between 0 and 7756 m [43]. Southwest China is the main distribution area of karst landscapes in China, with karst areas covering up to 550,000 km$^2$ [44]. Southwest China encompasses six distinct climatic zones, ranging from a tropical climate in the southern regions, through a subtropical climate in the central areas, to a plateau climate in the northwest [45]. This plateau climate zone, notable for its extensive grasslands, spans approximately 13% of Southwest China's total area. The predominant vegetation types across Southwest China are shrubs and forests, which together constitute about 67% of the region. Southwest China experiences marked seasonal variations, characterized by dry and wet periods [46]. Spring and winter bring low temperatures and minimal precipitation, while summer and autumn are marked by high temperatures and significant precipitation [47].

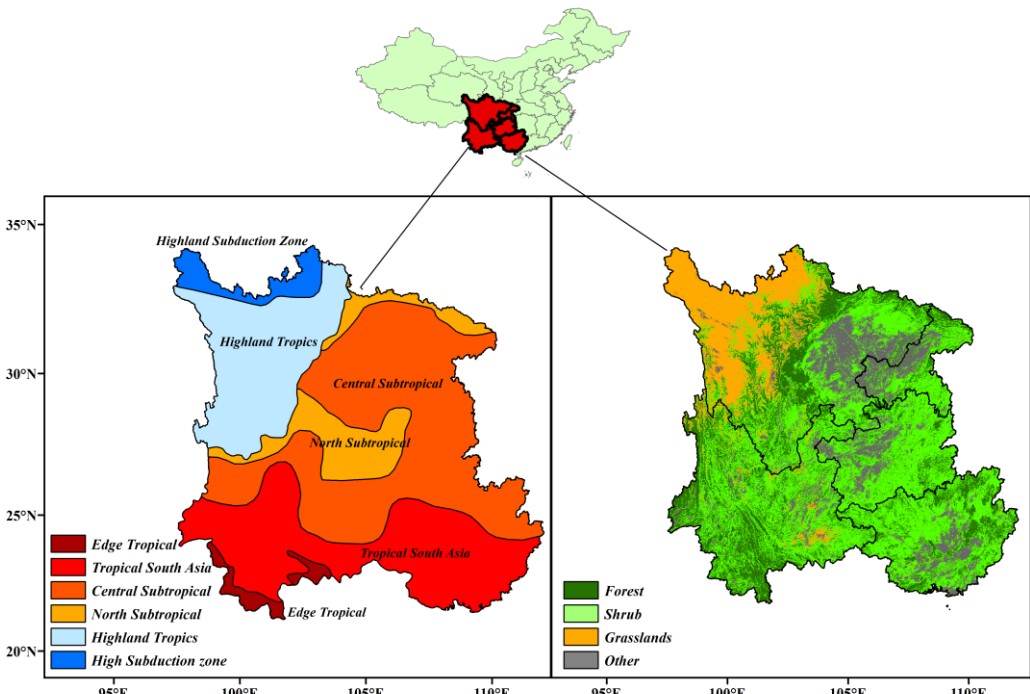

**Figure 1.** Map of elevation changes in the Southwest China. The distribution of climatic zones in the study area is shown on the left and the distribution of land use in the study area is shown on the right.

### 4. Results

#### 4.1. Evaluation of Simulation Results in Southwest CHINA

This study utilized soil moisture data from GLDAS, ERA5, and ESA-CCI remote sensing for validating the CLM4.5 model's simulated soil moisture in Southwest China. To enhance the validation process, Southwest China was segmented into four distinct regions based on administrative divisions: Yunnan, Sichuan and Chongqing, Guizhou, and Guangxi. This approach facilitated a targeted validation by integrating reanalysis and remote sensing data. Figure 2 illustrates a significant linear correlation between CLM4.5-simulated soil moisture and reanalyzed soil moisture data. Table 1 corroborates this, showing that the correlation between simulated soil moisture across various regions and reanalysis data exceeded 0.7, with all results passing the significance test ($p < 0.05$). Both the root mean square error (RMSE) and the mean absolute error (MAE) for the comparison of simulated and reanalyzed soil moisture were below 0.06 and 0.036, respectively. These

findings validate the effectiveness of the CLM4.5 model for simulating soil moisture in Southwest China.

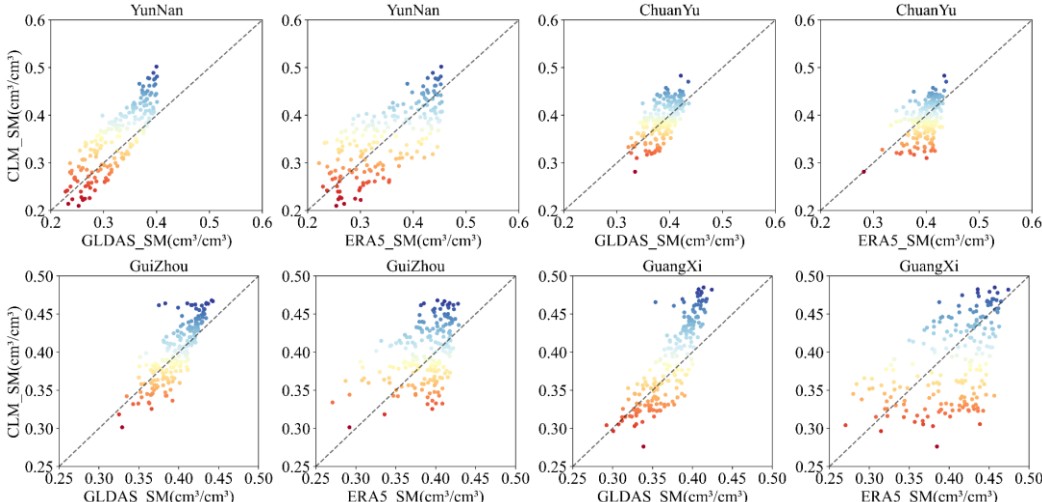

**Figure 2.** Comparison of soil moisture products of CLM4.5, GLDAS, and ERA5. Red represents low soil moisture and blue represents high soil moisture.

**Table 1.** R, RMSE, and MAE of soil moisture content of various products in Southwest China.

| Area | | R | RMSE | MAE |
|---|---|---|---|---|
| Yunnan | GLDAS | 0.843 * | 0.047 | 0.0014 |
| | ERA5 | 0.718 * | 0.051 | 0.0026 |
| Chuanyu | GLDAS | 0.719 * | 0.045 | 0.0024 |
| | ERA5 | 0.689 * | 0.032 | 0.0010 |
| Guizhou | GLDAS | 0.816 * | 0.040 | 0.0031 |
| | ERA5 | 0.70 * | 0.052 | 0.0027 |
| Guangxi | GLDAS | 0.86 * | 0.047 | 0.0036 |
| | ERA5 | 0.701 * | 0.052 | 0.0027 |

* The correlation passed the 95% significance test.

Figure 3 shows the scatter plots of CLM4.5-simulated soil moisture and ESA-CCI soil moisture in four regions in the southwest. Combined with Table 2, it can be found that the correlation between CLM4.5-simulated soil moisture and ESA-CCI soil moisture in all four regions passed the 95% significance test, with the correlation coefficient above 0.6. The RMSEs of model-simulated soil moisture and remotely sensed soil moisture were around 0.06, and the MAEs were below 0.05.

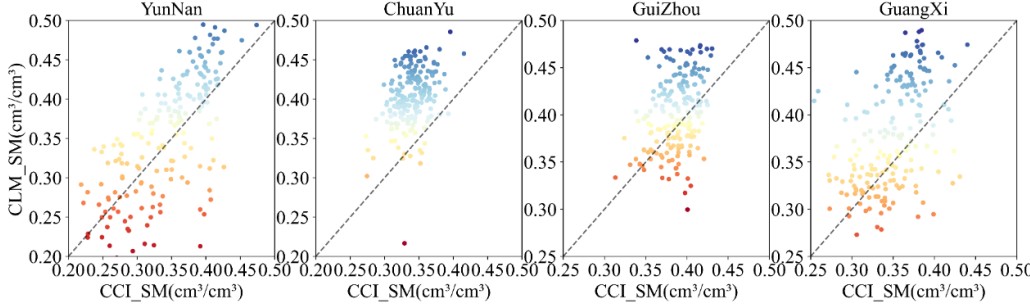

**Figure 3.** Comparison of soil moisture products of CLM4.5 and ESA-CCI. Red represents low soil moisture and blue represents high soil moisture.

**Table 2.** R, RMSE, and MAE of CLM4.5 simulated soil moisture and EAS-CCI soil moisture in Southwest China.

| Area | R | RMSE | MAE |
|---|---|---|---|
| Yunnan | 0.6811 * | 0.0572 | 0.0032 |
| Chuanyu | 0.6027 * | 0.0719 | 0.0049 |
| Guizhou | 0.7217 * | 0.0405 | 0.0016 |

* The correlation passed the 95% significance test.

By comparing the validation indexes of CLM4.5 simulation results with reanalysis data and remote sensing data, it can be found that the deviation between simulation results and reanalysis soil moisture in Southwest China is smaller. This experiment speculates that in addition to the model itself, the missing data of ESA-CCI itself may also be the reason for the obvious variability of the validation results. Taking the spatial distribution of monthly-scale ESA-CCI soil moisture in Southwest China from January to April 2005 (Figure 4) as an example, it can be found that there are also obvious missing data of ESA-CCI monthly soil moisture in Southwest China. In particular, the missing data of ESA-CCI soil moisture in the Sichuan and Chongqing regions are particularly serious, and considering the low correlation between CLM4.5-simulated soil moisture and ESA-CCI soil moisture in the Sichuan and Chongqing regions in Figure 4 as well as Table 2, the missing data may be the reason for the discrepancy in the validation results. Combined with the above validation results, we can conclude that the soil moisture data simulated by Opportunity CLM4.5 are in good agreement with the soil moisture products at this stage and can be used for further research.

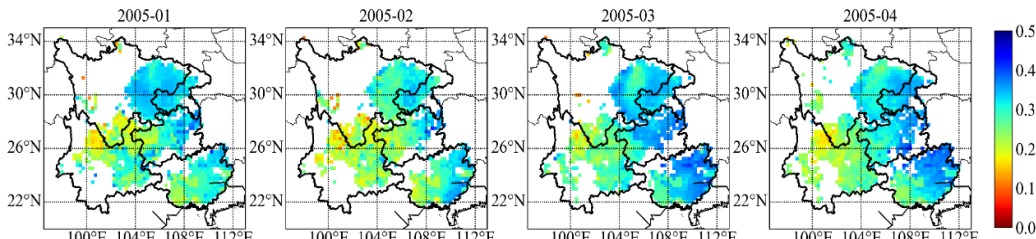

**Figure 4.** Spatial distribution of ESA-CCI soil moisture in Southwest China from January to April 2005. Red represents low soil moisture and blue represents high soil moisture.

### 4.2. Spatio-Temporal Correlation Analysis of VPD, SM, and SIFyield in Southwest China

To more effectively analyze the relationship among SM, VPD, and SIFyield, we calculated the spatial and temporal variations in the correlation between soil moisture (SM) or VPD and SIFyield across Southwest China. Figure 5 illustrates the change curves of average SIFyield, SM, and VPD in the southwest from 2005 to 2020. The changes in SIF yield coincide with those in SM, while the change in SIFyield shows an opposite trend to the change in VPD. For instance, the rise in SIF yield in 2012, 2014, and 2020 was accompanied by a rise in SM and a fall in VPD, whereas the fall in SIF yield in 2011, 2013, and 2019 was accompanied by a rise in VPD. The correlation coefficients of average SIFyield, SM, and VPD from 2005 to 2020 were calculated. It was found that the correlation coefficient between SIFyield and SM in Southwest China was 0.375 ($p < 0.05$), and the correlation coefficient between SIFyield and VPD was $-0.28$ ($p < 0.05$).

To examine the detailed spatial–temporal correlations among soil moisture, vapor pressure deficit, and SIFyield in Southwest China, we calculated the annual scale correlation coefficients for each parameter individually (Figure 6). Figure 6 reveals that, from 2005 to 2020, soil moisture (SM) and SIFyield exhibited a significant positive correlation in over 78% of the vegetation ecological zones in Southwest China, with a correlation coefficient of around 0.4 ($p < 0.05$). Conversely, the vapor pressure deficit (VPD) and SIFyield were negatively correlated in more than 80% of the areas, showing a correlation coefficient of

approximately −0.5. Analysis by vegetation type indicates that SM and SIFyield correlations are predominantly positive in shrub and forest ecosystems, whereas grasslands show a negative correlation with SIFyield. Additionally, the spatial and temporal correlation patterns between VPD and SIFyield display a clear contrast to those observed between SM and SIFyield.

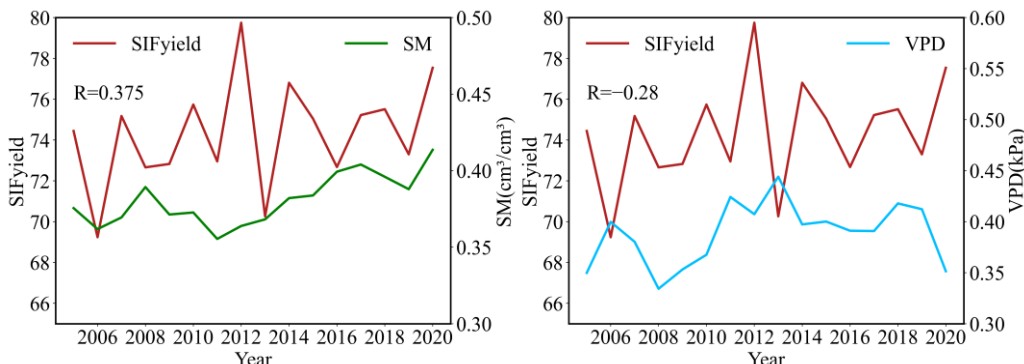

**Figure 5.** Annual scale variation of soil moisture, VPD, and SIFyield in Southwest China from 2005 to 2020.

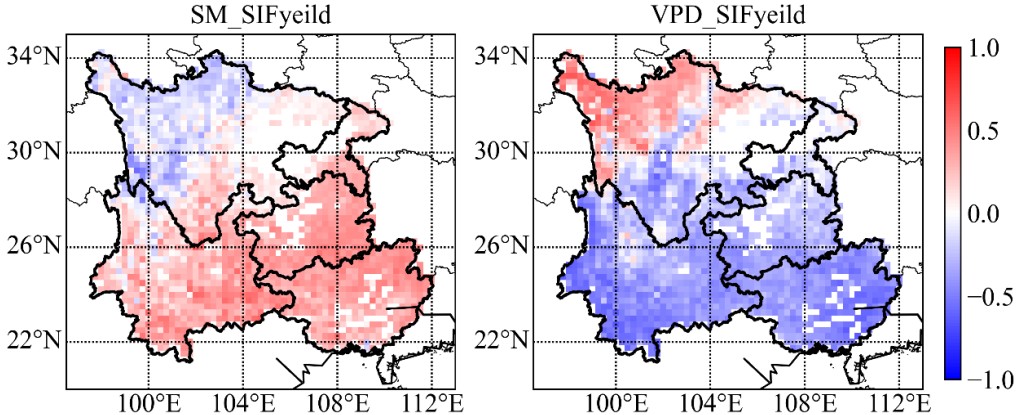

**Figure 6.** Spatio-temporal correlation between SM, VPD, and SIFyield in Southwest China. Red represents positive correlation, blue represents negative correlation, and white represents non-vegetated areas.

A strong coupling exists among VPD, SM, and SIFyield within vegetation ecosystems in Southwest China. To more effectively discern the individual impacts of SM and VPD on SIFyield, it is essential to decouple their strong interrelationship. Figure 7 illustrates a notable decline in the correlation between soil moisture (SM) and vapor pressure deficit (VPD) as the temporal resolution increases. On an annual scale, the correlation coefficient in Southwest China is approximately −0.53, decreasing to about −0.43 on a monthly scale, and further to around −0.39 on an 8-day scale. Despite the significant reduction in correlation coefficients with the increasing temporal scale, a certain level of coupling between SM and VPD persists even at the 8-day scale. To mitigate the correlation between the two drivers, a data binning approach was employed on SM and VPD at the 8-day scale. The correlation coefficient between SM and VPD dropped to about −0.03 in the binning of SM and about −0.07 in the data binning of VPD. This result indicates that the data binning can effectively decouple SM and VPD. Consequently, in subsequent experiments, we used a data binning approach and Formulas (2) and (3) in conjunction with an 8-day scale dataset to calculate the respective abilities of SM and VPD to influence regional ecosystems.

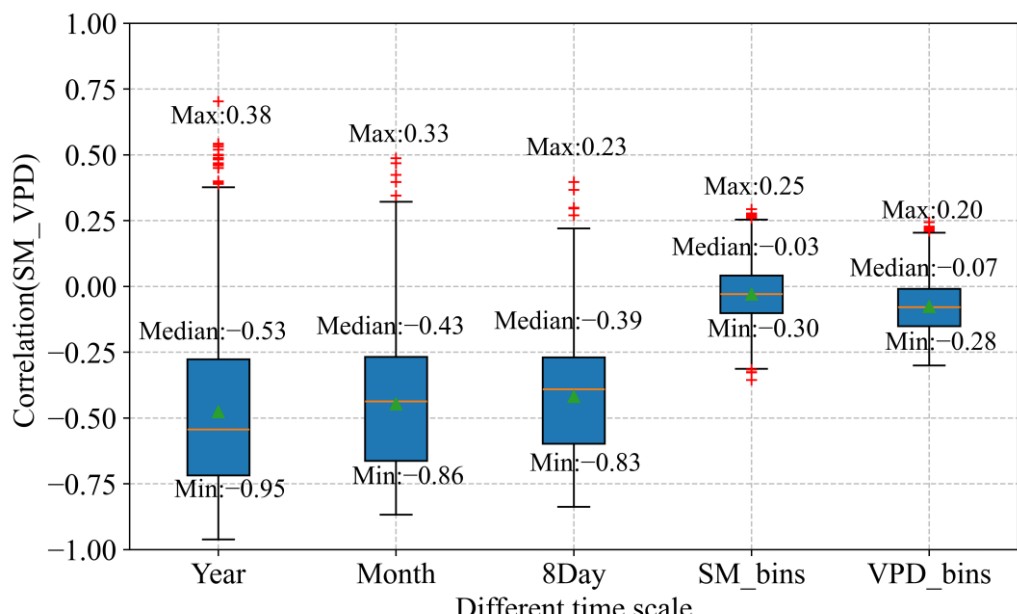

**Figure 7.** Correlation between SM and VPD at different time scales in Southwest China. The yellow line represents the median of the correlation and the green triangle represents the mean of the correlation. The red color is the outliers.

### 4.3. Analysis of the Respective Impact of SM and VPD on SIFyields

Following the data binning process, we utilized Equations (2) and (3) to calculate the individual effects of SM and VPD on SIFyield in Southwest China. To obtain more spatially comparable results, we normalized the calculation on an image-by-image basis using the image value at 90% of its position. Figure 8 reveals that the increase in SIFyield, attributed to the influence of low SM, predominantly occurs in the southwestern part of the study area, where forests and shrubs are the primary vegetation types. Conversely, areas experiencing a decline in SIFyield due to low SM encompass a larger portion of the study region. The fluctuation in SIFyield induced by low SM ranges from −0.3 to 0.3. Approximately 79% of the southwestern vegetated ecosystem experienced a decrease in SIFyield attributed to low SM, while about 21% of the area saw an increase in SIFyield under similar conditions of low SM.

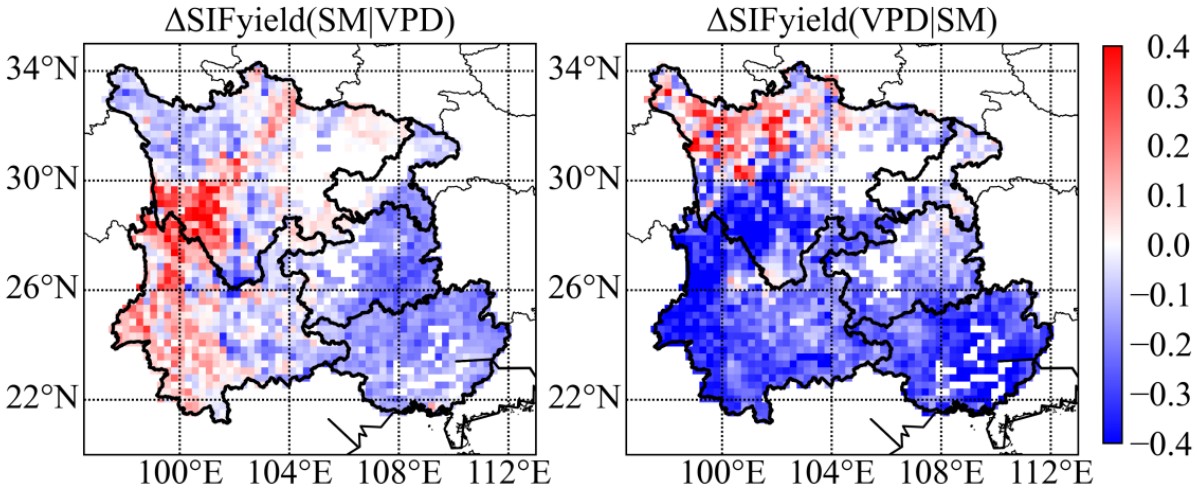

**Figure 8.** Effect of low SM and high VPD on SIFyield, respectively. ΔSIFyield (SM | VPD) is the effect of low SM on SIFyield, and ΔSIFyield (VPD | SM) is the effect of high VPD on SIFyield.

In Southwest China, vegetation ecosystems primarily exhibit a declining SIFyield trend in response to high VPD. An analysis of land use classification in the southwest reveals that grasslands, which constitute approximately 19% of the region's area, mainly show increases in SIFyield due to high VPD. Conversely, areas experiencing decreases in SIFyield due to high VPD predominantly feature forests and shrubs. On the whole, the ratio of areas with increased versus decreased SIFyield attributable to high VPD is about 1 to 4. The variation in SIFyield associated with high VPD, ranging between −0.4 and 0.4, is significantly more pronounced than that caused by low soil moisture.

To quantify and visualize the differences observed in this experiment, we used the absolute values of the changes in SIFyield resulting from low SM and high VPD, as depicted in Figure 9. This figure demonstrates that the change in SIFyield caused by high VPD is more significant than that caused by low SM. Specifically, ΔSIFyield (SM | VPD) was less than ΔSIFyield (VPD | SM) in approximately 75% of the vegetated ecosystems in Southwest China, whereas around 25% of the areas showed more significant effects caused by SM, primarily involving shrubs and some grassland vegetation types. Overall, these findings suggest that VPD is more critical than SM for ecosystem production in Southwest China.

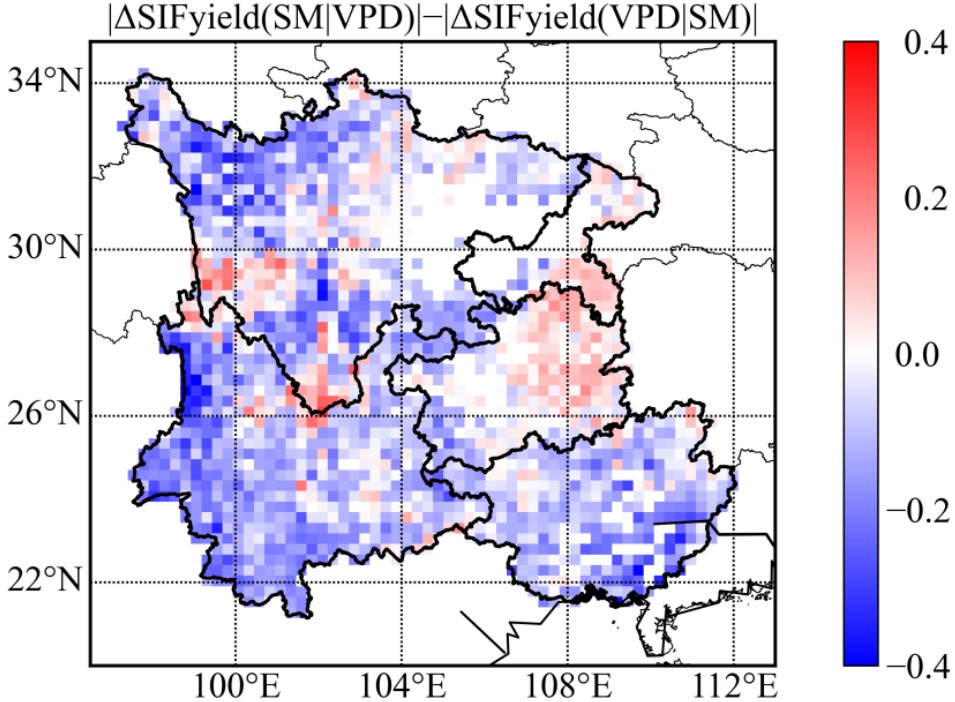

**Figure 9.** Absolute differences in the impact of SM and VPD on SIFyield in Southwest China.

The data binning can decouple the correlation between SM and VPD more effectively. In order to better study the relationship between SM, VPD, and SIFyield, we assessed the impact of VPD on the correlation strength between SM and SIFyield, as well as the influence of SM on the correlation strength between VPD and SIFyield. The outcomes of this analysis are presented in Figures 10 and 11. Figure 10 provides a clear visual representation of the correlation coefficient between VPD and SIFyield, which decreases gradually from −0.5 to −0.15, with an increase in SM from 10% to approximately 40%. However, the change in correlation is negligible as SM increases from 50% to about 80%. Notably, as SM reaches its maximum value of 100%, the correlation coefficient between VPD and SIFyield shows a moderate increase, albeit with a smaller magnitude. This result implies that the process of increasing the soil moisture from extremely low to a normal state significantly affects the SIFyield, while the increase in soil moisture in a normal state affects the SIFyield to a very small extent.

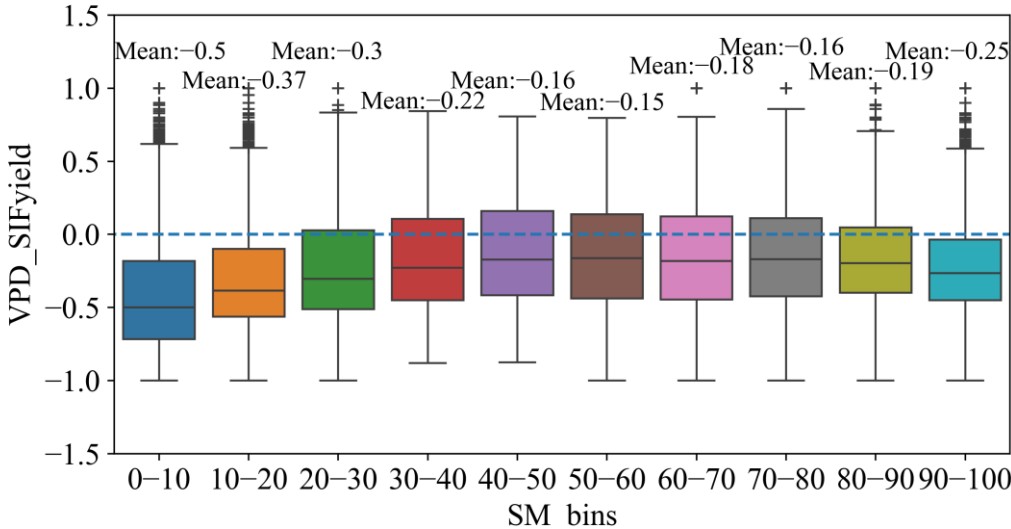

**Figure 10.** Variation in the correlation between VPD and SIFyield under different SM bins in Southwest China. The black line in the box plot represents the median correlation.

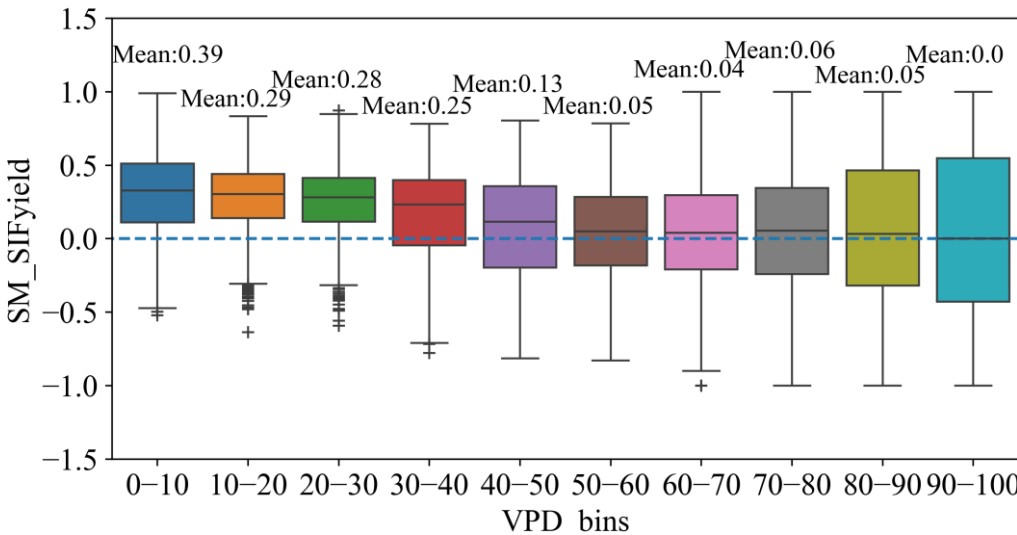

**Figure 11.** Changes in correlation between SM and SIFyield under different VPD bins in Southwest China. The black line in the box plot represents the median correlation.

As depicted in Figure 11, VPD significantly affects the correlation between SM and SIFyield. Specifically, when VPD increases from 10% to approximately 50%, the correlation coefficient between SM and SIFyield drops from 0.4 to about 0.2. The correlation between SM and SIFyield remains relatively stable as VPD ranges from 50% to 70%. However, a marked variability in the correlation coefficients between SM and VPD is observed when VPD increases from 70% to 100%. Notably, within the 90–100% VPD bin, the correlation coefficients between SM and SIFyield in Southwest China exhibit significant fluctuations, ranging from −1 to 1, a variance considerably greater than in other VPD data bins. This finding indicates that the variation in SIFyield caused by high VPD is significantly greater than the variation caused by low SM.

To thoroughly examine the impacts of SM and VPD on the variations in SIFyield across various ecosystems in Southwest China, we utilized MODIS land use classification data. This approach enabled us to investigate the effects of SM and VPD on SIFyield within the specific contexts of forest, shrub, and grassland vegetation types. The findings, illustrated in Figure 12, reveal that the correlation between VPD and SIFyield within woodland ecosystems exhibited a trend that initially decreased and then increased with

a rising number of SM bins. Conversely, for shrub and grassland vegetation types, the correlation between VPD and SIFyield demonstrated an increasing and then decreasing trend as the number of SM bins increased.

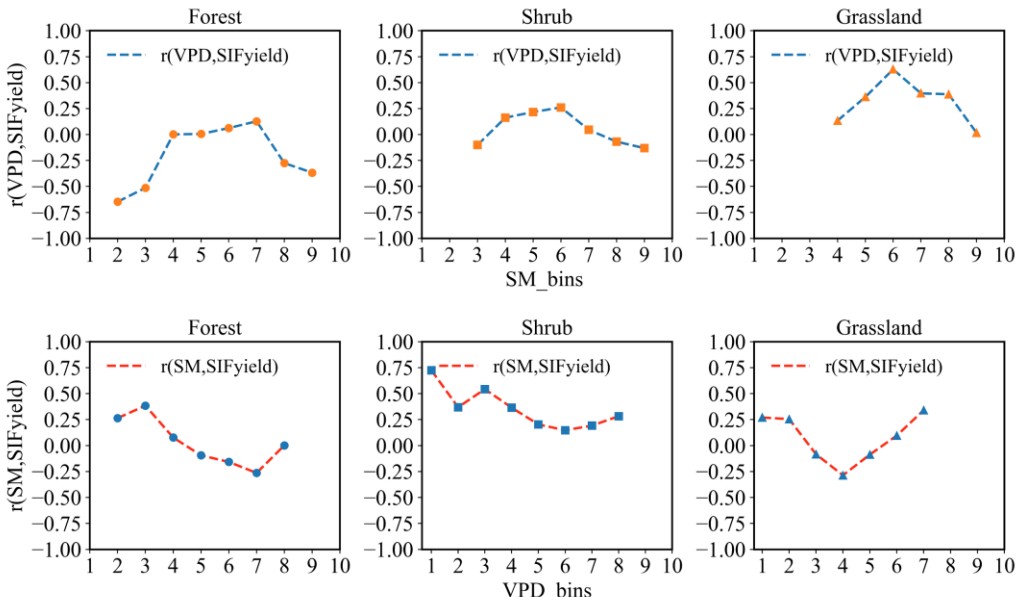

**Figure 12.** Changes in SM_SIFyield correlation by VPD and VPD_SIFyield correlation by SM under different vegetation systems in Southwest China. The orange color represents the change in VPD and SIFyield correlation under different SM bins. The blue color represents the variation in SM and SIFyield correlation under different VPD bins. When the amount of data in the data bins of SM and VPD is less than 50, the results of that partition are excluded.

With an increasing number of VPD bins, the correlation between SM and SIFyield tended to decrease and then increase under the forest type. The correlation between SM and SIFyield showed highly significant decreasing trends with increasing VPD under shrub type. The effect of VPD on the correlation between SM and SIFyield was more complex under the grassland type. In order to gain a deeper insight into the effect of VPD on the correlation between SM and SIFyield, we calculated the slope of the linear regression line of the SIFyield correlation curve for each vegetation type and used the magnitude of the slope as a measure of this effect. Our results, presented in Table 3, show that the effect of VPD on the correlation between SM and SIFyield was significantly greater than the effect of SM on VPD. This phenomenon was particularly evident in forest and shrub ecosystems, where the difference between the two effects on the correlation with each other and with SIFyield was most pronounced. In grassland ecosystems, the impact of SM on VPD and SIFyield was found to be similar to, yet slightly greater than, the impact of VPD on SM and SIFyield.

**Table 3.** Statistics on the trend of correlation between VPD, SM, and SIFyield for different vegetation types, as influenced by another element.

|  | Forest | Shrub | Grassland |
|---|---|---|---|
| VPD_bins | 0.083 | 0.063 | 0.018 |
| SM_bins | 0.043 | 0.026 | 0.024 |

## 5. Discussion and Conclusions

This study utilized the CLM4.5 surface process model, integrated with GLDAS atmospheric data, to simulate soil moisture and additional variables in Southwest China from 2005 to 2020. Employing data binning and other analytical techniques, we quantitatively

evaluated the impacts of soil moisture and vapor pressure deficit on ecosystem productivity within the region. Furthermore, the interplay between soil moisture, vapor pressure deficit, and vegetation productivity was examined across various land use types, incorporating land use data. The analysis led to several key conclusions:

(1) In Southwest China, the spatial and temporal distributions of the correlation coefficients between soil moisture and SIFyield, and between vapor pressure deficit and SIFyield, were largely inversely proportional. The ratio of positive to negative correlations between SM and SIFyield was approximately 4:1. Specifically, in forest and shrub ecosystems, a positive correlation was observed between SM and SIFyield. Conversely, in grassland ecosystems, a negative correlation was found between SM and SIFyield.

(2) The correlation between SM and VPD decreased significantly as the time scale of the data increased. The combination of the 8-day-scale data and the data binning method reduced the correlation between SM and VPD to about 0.03, which effectively decoupled SM and VPD.

(3) In Southwest China, high VPD significantly influenced vegetation productivity in approximately 75% of vegetated areas more than low SM did. Forests and shrubs experienced greater impacts from high VPD, whereas grassland ecosystems were more susceptible to the effects of low SM.

In this study, we meticulously explored and examined the criticality of the influence of low SM and high VPD on regional ecosystem productivity, utilizing a comprehensive set of soil moisture, vapor pressure deficit, and SIF data. This finding unequivocally demonstrated that high VPD had a more substantial impact on the production of ecosystems in Southwest China than low SM. Further scrutinizing the data with respect to various vegetation types in the region, the effect of VPD variations on the correlation between SM and SIFyield was found to be significantly more prominent, particularly in the forest and shrub vegetation types. Conversely, the influence of SM on the correlation between VPD and SIFyield was more potent under the grassland type. These results confirm that VPD has a significantly higher influence on regional ecosystem production than SM.

This study introduces several innovations compared to previous research. Firstly, it synthesizes the methodologies of prior studies and incorporates temperature, vapor pressure deficit, PAR, and FPAR data. This approach significantly reduces the influence of temperature, solar radiation, and precipitation, thereby enhancing the accuracy of our findings. Secondly, we utilized the CLM4.5 model, a highly sophisticated land surface process model, to accurately simulate regional hydrological variables. This model enables the generation of temporally and spatially continuous soil moisture data across Southwest China over an extended period. The study not only confirms the applicability of the CLM4.5 model in Southwest China, but also provides a comprehensive set of soil moisture and VPD data. Finally, by quantitatively analyzing the impact of changes in SM_bins on VPD and SIFyield, as well as the impact of changes in VPD_bins on SM and SIFyield, we reaffirm the critical influence of high VPD on vegetation productivity in Southwest China. Incorporating land use data, we further examine and discuss the variability of these impacts across different vegetation types. Our findings indicate that soil moisture has a more pronounced effect on grassland productivity, whereas VPD more significantly affects the productivity of shrubs and forests.

This study also acknowledges certain limitations. Primarily, the constraints inherent to the GLDAS atmospheric data led to a lower spatial resolution in the CLM4.5 simulated data. This limitation restricts our capacity to acquire more detailed spatial information. Moreover, this study primarily concentrates on contrasting the impacts of VPD and SM on regional productivity, employing various methods to minimize the influences of temperature, radiation, and other factors wherever feasible. However, it is important to acknowledge that completely eliminating the effects of these elements may not be entirely achievable, which could influence our findings to some extent. Hence, future research might incorporate different scenario simulations to more effectively isolate and reduce the impacts of these additional factors.

**Author Contributions:** Conceptualization, X.H.; data curation, X.S. and M.M.; formal analysis, Y.X. and J.W.; funding acquisition, M.M. and X.H.; methodology, X.S. and M.Z.; project administration.; software, X.S.; supervision, M.M. and X.H.; validation, Y.X., Z.S. and M.M.; visualization, X.S. and X.H.; writing—original draft, X.S.; writing—review and editing, J.W. and X.H. All authors have read and agreed to the published version of the manuscript.

**Funding:** This work is supported by the special fund for youth team of Southwest University project (grant numbers: SWU-XJLJ202305) and the Chinese High-resolution Earth Observation System of China (project number: 21-Y20B01-9001-19/22). Xujun Han: SWU-XJLJ202305; Mingguo Ma: 21-Y20B01-9001-19/22.

**Data Availability Statement:** No new data were created or analyzed in this study. Data sharing is not applicable to this article.

**Acknowledgments:** In this study, the atmospheric drive data used came from the Global Land Data Assimilation System (GLDAS Noah Land Surface Model L4 3 hourly 0.25 × 0.25-degree V2.1 Greenbelt), and the statistical year data used came from the Chinese Government Statistical Yearbook. Comparison of soil moisture data from GLDAS with ERA5. We sincerely thank the anonymous reviewers.

**Conflicts of Interest:** The authors declare no conflicts of interest.

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
