# Peer review of "The Respective Effects of Vapor Pressure Deficit and Soil Moisture on Ecosystem Productivity in Southwest China"

_remotesensing, doi:10.3390/rs16081316_

Round 1

Reviewer 1 Report

Comments and Suggestions for Authors

This study explores the individual and combined effects of soil moisture and vapour pressure deficits on ecosystem productivity in the Southwest. By reading the manuscript, I have the following questions and comments:

1.     In the introductory section, the author lists a large number of relevant studies, but the summary and overview of these studies is inadequate; the presentation of the necessity, innovativeness and significance of the study needs to be improved;

2.     Abbreviations should be added in full where they first appear in the manuscript (e.g., lines 98-104) so that the reader can read and understand them;

3.     Line109-Line113, the appropriate theoretical support or reference should be added;

4.     Figure 1, the authors are advised to use standard maps. It is recommended that Figure 2 be integrated with Table 1; furthermore, according to Table 1, the goodness of fit (R2) of the two data sets is low and does not effectively illustrate a synergistic relationship between them;

5.     The discussion section of the study is inadequate. Extended analyses of the findings as well as content on the uniqueness and significance of the study need to be added;

In summary, it is recommended that the manuscript should be given a major revision.

Author Response

1 In the introductory section, the author lists a large number of relevant studies, but the summary and overview of these studies is inadequate; the presentation of the necessity, innovativeness and significance of the study needs to be improved

Re: Thank you very much for your suggestions, I have made changes to the introductory section. And we have re-summarized and re-explained the innovation section.

  1. Abbreviations should be added in full where they first appear in the manuscript (e.g., lines 98-104) so that the reader can read and understand them

Re: Thanks for the heads up, I've added the full name to the first occurrence of the abbreviation.

  1. Line109-Line113, the appropriate theoretical support or reference should be added

Re: Thanks for the heads up, I have added relevant references.

  1. Figure 1, the authors are advised to use standard maps. It is recommended that Figure 2 be integrated with Table 1; furthermore, according to Table 1, the goodness of fit (R2) of the two data sets is low and does not effectively illustrate a synergistic relationship between them;

Re: Thank you very much for your suggestion, we modified the map. For your question about the smaller R2, I make the following answers: (1) The simulation of soil moisture in the Southwest region using CLM4.5 has the issue of underestimating and overestimating soil moisture, which is also present in most of the analyzed data at this stage. (2) R2 alone is not a sufficient assessment indicator and may lead to inaccurate results. Therefore, we also used RMSE and MAE. The combined results of the three indicators show that the data simulated by CLM4.5 are more consistent with GLDAS and ERA5.

5 The discussion section of the study is inadequate. Extended analyses of the findings as well as content on the uniqueness and significance of the study need to be added

Re: Thank you very much for your suggestion that we add an extended analysis of the findings as well as content on the uniqueness and significance of the study, the results of which are shown below.

“This study utilized the CLM4.5 surface process model, integrated with GLDAS atmospheric data, to simulate soil moisture and additional variables in Southwest China from 2005 to 2020. Employing data binning and other analytical techniques, we quantitatively evaluated the impacts of soil moisture and vapor pressure deficit on ecosystem productivity within the region. Furthermore, the interplay between soil moisture, vapor pressure deficit, and vegetation productivity was examined across various land use types, incorporating land use data. The analysis led to several key conclusions:

(1) In Southwest China, the spatial and temporal distributions of the correlation coefficients between soil moisture and SIFyield, and between vapor pressure deficit and SIFyield, were largely inversely proportional. The ratio of positive to negative cor-relations between SM and SIFyield was approximately 4:1. Specifically, in forest and shrub ecosystems, a positive correlation was observed between SM and SIFyield. Con-versely, in grassland ecosystems, a negative correlation was found between SM and SIFyield.

(2) The correlation between SM and VPD decreased significantly as the time scale of the data increased. The combination of the 8-day scale data and the data binning method can reduce the correlation between SM and VPD to about 0.03, which can ef-fectively decouple SM and VPD.

(3) In Southwest China, high VPD significantly influenced vegetation productivity in approximately 75% of vegetated areas more than low SM did. Forests and shrubs experienced greater impacts from high VPD, whereas grassland ecosystems were more susceptible to the effects of low SM.

In this study, we meticulously explored and examined the criticality of the influence of low SM and high VPD on regional ecosystem productivity, utilizing a comprehensive set of soil moisture, vapor pressure deficit, and SIF data. This finding unequivocally demonstrated that high VPD had a more substantial impact on the production of ecosystems in Southwest China than low SM. Further scrutinizing the data with respect to various vegetation types in the region, the effect of VPD variations on the correlation between SM and SIFyield was found to be significantly more prominent, particularly in the forest and shrub vegetation types. Conversely, the influence of SM on the correlation between VPD and SIFyield was more potent under grassland types. These results confirm that VPD has a significantly higher influence on regional ecosys-tem production than SM.

This study introduces several innovations compared to previous research. Firstly, it synthesizes the methodologies of prior studies and incorporates temperature, vapor pressure deficit, PAR, and FPAR data. This approach significantly reduces the influ-ence of temperature, solar radiation, and precipitation, thereby enhancing the accura-cy of our findings. Secondly, we utilize the CLM4.5 model, a highly sophisticated land surface process model, to accurately simulate regional hydrological variables. This model enables the generation of temporally and spatially continuous soil moisture data across Southwest China over an extended period. The study not only confirms the applicability of the CLM4.5 model in Southwest China but also provides a comprehensive set of soil moisture and VPD data. Finally, by quantitatively analyzing the impact of changes in SM_bins on VPD and SIFyield, as well as the impact of changes in VPD_bins on SM and SIFyield, we reaffirm the critical influence of high VPD on vege-tation productivity in Southwest China. Incorporating land use data, we further ex-amine and discuss the variability of these impacts across different vegetation types. Our findings indicate that soil moisture has a more pronounced effect on grassland productivity, whereas VPD more significantly affects the productivity of shrubs and forests.

This study also acknowledges certain limitations. Primarily, the constraints inherent to the GLDAS atmospheric data have led to a lower spatial resolution in the CLM4.5 simulated data. This limitation restricts our capacity to acquire more detailed spatial information. Moreover, this study primarily concentrated on contrasting the impacts of VPD and SM on regional productivity, employing various methods to minimize the influences of temperature, radiation, and other factors wherever feasible. However, it is important to acknowledge that completely eliminating the effects of these elements may not be entirely achievable, which could influence our findings to some extent. Hence, future research might incorporate different scenario simulations to more effectively isolate and reduce the impacts of these additional factors.”

Reviewer 2 Report

Comments and Suggestions for Authors

This study adds to the field by using the CLM4.5 model to simulate soil moisture and vapour pressure deficits and provide a detailed analysis of their effects on ecosystem productivity in southwest China. It advances the understanding of the relative influence of these factors on different vegetation types, pointing out that VPD is more influential in 70% of the study area, while MS has a greater influence in the remaining 30%. The use of SIFYield as a productivity indicator and the binary data approach to separate the interaction between MS and VPD at finer temporal resolutions are novel contributions that increase the granularity of the analysis. This research differs from other studies by quantifying the coupling effect of MS and VPD, which reduces with increasing temporal data resolution.

Major remarks:

1. Line 331 – why the temperature is not considered? I kindly ask the authors to justify this limitation of their study. To enhance the study, the authors should consider controlling additional environmental variables, such as temperature, precipitation, and radiation, as these can significantly influence ecosystem productivity. The study could include controls for other environmental factors, such as temperature and precipitation, to isolate the effects of soil moisture deficit and vapour pressure on ecosystem productivity. But I repeat, I do not understand why the authors left this rather important limitation of the study.

2. Figure 2: I'm not sure it's clear what the colors from red to blue mean. I ask the authors to add some comments here.

3. Figure 3 should be described in more detail and redesigned. Basically, it is difficult to follow the correlation factors mentioned by the authors in the text. I don't understand why the authors here didn't prefer to represent correlation graphs, especially since they don't refer to the year in the discussion of figure 3.

4. The authors should bring more discussion about the results presented in most of the figures. For example, why they preferred to highlight the maximum values of certain parameters. Wider discussions should be held.

5. I recommend to the authors that the annex be included in the text of the paper.

6. Chapters 5 and 6 would be better merged under the title "Discussions and Conclusions"

Author Response

Major remarks:

  1. Line 331 – why the temperature is not considered? I kindly ask the authors to justify this limitation of their study. To enhance the study, the authors should consider controlling additional environmental variables, such as temperature, precipitation, and radiation, as these can significantly influence ecosystem productivity. The study could include controls for other environmental factors, such as temperature and precipitation, to isolate the effects of soil moisture deficit and vapor pressure on ecosystem productivity. But I repeat, I do not understand why the authors left this rather important limitation of the study.

Re: Thank you for your advice. Regarding your questions, temperature was not entirely excluded from this study. To eliminate other meteorological drivers that may have a significant impact on the results, we screened the data as follows: for each pixel, we analyzed only the data from the following dates: (1) daily mean temperature >15 °C; (2) daily mean VPD >0.5 kPa. To limit the data to relatively warm and sunny dates, we used temperature and VPD. Additionally, we calculated the SIFyield to exclude the mixed effects of radiation on vegetation photosynthesis, specifically excluding the effects of PAR and FPAR. It should be noted that temperature is the primary factor affecting vegetation productivity. Although we were able to mitigate the impact of certain factors through the aforementioned procedures, we were unable to completely eliminate the influence of temperature, rainfall, and other variables. This limitation is acknowledged in our study. We have included a detailed description of our methodology in the corresponding section and revised the limitations section to enhance reader comprehension.

  1. Figure 2: I'm not sure it's clear what the colors from red to blue mean. I ask the authors to add some comments here.

Re: Thank you very much for your suggestion, I have described the color part of figure 2.

Figure 2. Comparison of soil moisture products of CLM4.5, GLDAS and ERA5. Red represents low soil moisture and blue represents high soil moisture.

  1. Figure 3 should be described in more detail and redesigned. Basically, it is difficult to follow the correlation factors mentioned by the authors in the text. I don't understand why the authors here didn't prefer to represent correlation graphs, especially since they don't refer to the year in the discussion of figure 3.

Re: Thank you very much for your suggestion, I have redesigned Figure 3 and added a description of the relevant aspects.

 “To more effectively analyze the relationship among SM, VPD, and SIFyield, we calculated the spatial and temporal variations in the correlation between Soil Moisture (SM) or VPD and SIFyield across Southwest China. Figure 5 illustrates the change curves of average SIFyield, SM and VPD in the southwest from 2005 to 2020. The changes in SIF yield coincide with those in SM, while the change in SIFyield shows an opposite trend to the change in VPD. For instance, the rise in SIF yield in 2012, 2014, and 2020 is accompanied by a rise in SM and a fall in VPD, whereas the fall in SIF yield in 2011, 2013, and 2019 is accompanied by a rise in VPD. The correlation coefficients of average SIFyield, SM, and VPD from 2005 to 2020 were calculated. It was found that the correlation coefficient between SIFyield and SM in Southwest China was 0.375 (p<0.05), and the correlation coefficient between SIFyield and VPD was -0.28 (p<0.05).”

Figure 5. Annual scale variation of soil moisture, VPD and SIFyield in Southwest China from 2005 to 2020

  1. The authors should bring more discussion about the results presented in most of the figures. For example, why they preferred to highlight the maximum values of certain parameters. Wider discussions should be held.

Re: Thank you very much for your suggestion, I have re-discussed and summarized the variations of the parameters in the relevant diagrams.

  1. I recommend to the authors that the annex be included in the text of the paper.

Re: Thank you very much for your suggestion, I have added the results from the appendix into the text.

  1. Chapters 5 and 6 would be better merged under the title "Discussions and Conclusions"

Re: Thank you very much for your advice, I took your advice and made the changes.

“5. Discussion and Conclusions

This study utilized the CLM4.5 surface process model, integrated with GLDAS atmospheric data, to simulate soil moisture and additional variables in Southwest China from 2005 to 2020. Employing data binning and other analytical techniques, we quantitatively evaluated the impacts of soil moisture and vapor pressure deficit on ecosystem productivity within the region. Furthermore, the interplay between soil moisture, vapor pressure deficit, and vegetation productivity was examined across various land use types, incorporating land use data. The analysis led to several key conclusions:

(1) In Southwest China, the spatial and temporal distributions of the correlation coefficients between soil moisture and SIFyield, and between vapor pressure deficit and SIFyield, were largely inversely proportional. The ratio of positive to negative cor-relations between SM and SIFyield was approximately 4:1. Specifically, in forest and shrub ecosystems, a positive correlation was observed between SM and SIFyield. Conversely, in grassland ecosystems, a negative correlation was found between SM and SIFyield.

(2) The correlation between SM and VPD decreased significantly as the time scale of the data increased. The combination of the 8-day scale data and the data binning method can reduce the correlation between SM and VPD to about 0.03, which can ef-fectively decouple SM and VPD.

(3) In Southwest China, high VPD significantly influenced vegetation productivity in approximately 75% of vegetated areas more than low SM did. Forests and shrubs experienced greater impacts from high VPD, whereas grassland ecosystems were more susceptible to the effects of low SM.

In this study, we meticulously explored and examined the criticality of the influ-ence of low SM and high VPD on regional ecosystem productivity, utilizing a compre-hensive set of soil moisture, vapor pressure deficit, and SIF data. This finding une-quivocally demonstrated that high VPD had a more substantial impact on the produc-tion of ecosystems in Southwest China than low SM. Further scrutinizing the data with respect to various vegetation types in the region, the effect of VPD variations on the correlation between SM and SIFyield was found to be significantly more prominent, particularly in the forest and shrub vegetation types. Conversely, the influence of SM on the correlation between VPD and SIFyield was more potent under grassland types. These results confirm that VPD has a significantly higher influence on regional ecosys-tem production than SM.

This study introduces several innovations compared to previous research. Firstly, it synthesizes the methodologies of prior studies and incorporates temperature, vapor pressure deficit, PAR, and FPAR data. This approach significantly reduces the influ-ence of temperature, solar radiation, and precipitation, thereby enhancing the accura-cy of our findings. Secondly, we utilize the CLM4.5 model, a highly sophisticated land surface process model, to accurately simulate regional hydrological variables. This model enables the generation of temporally and spatially continuous soil moisture data across Southwest China over an extended period. The study not only confirms the ap-plicability of the CLM4.5 model in Southwest China but also provides a comprehensive set of soil moisture and VPD data. Finally, by quantitatively analyzing the impact of changes in SM_bins on VPD and SIFyield, as well as the impact of changes in VPD_bins on SM and SIFyield, we reaffirm the critical influence of high VPD on vege-tation productivity in Southwest China. Incorporating land use data, we further ex-amine and discuss the variability of these impacts across different vegetation types. Our findings indicate that soil moisture has a more pronounced effect on grassland productivity, whereas VPD more significantly affects the productivity of shrubs and forests.

This study also acknowledges certain limitations. Primarily, the constraints in-herent to the GLDAS atmospheric data have led to a lower spatial resolution in the CLM4.5 simulated data. This limitation restricts our capacity to acquire more detailed spatial information. Moreover, this study primarily concentrated on contrasting the impacts of VPD and SM on regional productivity, employing various methods to minimize the influences of temperature, radiation, and other factors wherever feasible. However, it is important to acknowledge that completely eliminating the effects of these elements may not be entirely achievable, which could influence our findings to some extent. Hence, future research might incorporate different scenario simulations to more effectively isolate and reduce the impacts of these additional factors.”

Reviewer 3 Report

Comments and Suggestions for Authors

Because of climate change with extreme hot weather, this research is very important area. To improve your research, I would like to recommend and ask some questions as follow:

1)    In figure 2, 2nd bottom right about GLADS_SM of GuanXi and upper right of ERA5_SM of Chuan Yu, linearity are different from other area. Do you explain why?

2)    In figure 3, SIFyield, there are year variation. Do you check the relationship between SIFyield and PAR or other parameters?

3)    Unfortunately, for me, it is not clearly about your explanations about the value of binning in Figure 5 and others. Are Yearly, monthly and 8 days SM or VPD?   Difficult to understand the relationship between SM_bins and/or VPD_Bins and year, month and 8 days.

4)    Can you also explain how to process binning? After figure 6 (line 252-271), can you explain the data binning can decouple the correlation between VPD and SIFyield? are figure 8 and figure 9 explained the decoupling? If so, I suggest you to add figure between VPD_SYFYield and SM.

5)    In figure 9, the variation of VPD_Bins (from 70-80, 80-90, 90-100) be big. Do you explain why?

6)    In figure 10, is it only one point information. If so, you should add other area information.

7)    In line 323, you mention drought mechanisms, but there is no clear information about drought mechanisms and relationships with VPD and SM. And, explain drought mechanisms, it is also important to study the relationship between SM/VPD and temperature, PAR and rainfall as well as LULCC.

8)    In line 341, you should spell out CLM4.5 and explain why you adopt CLM4.5.

Comments on the Quality of English Language

No comments. Suggest to do only spell out some of text. 

Author Response

Because of climate change with extreme hot weather, this research is very important area. To improve your research, I would like to recommend and ask some questions as follow:

1) In figure 2, 2nd bottom right about GLADS_SM of GuanXi and upper right of ERA5_SM of Chuan Yu, linearity are different from other area. Do you explain why?

Re: Thank you for your question and my answer is as follows:

In this study, the atmospheric data from GLDAS combined with the CLM model were used to simulate soil moisture data in southwest China. Figure 2 shows the results of CLM simulated soil moisture (0-10 cm) compared with soil moisture from GLDAS (0-10 cm) and ERA5 (0-7 cm). The CLM simulation results are more consistent with the GLDAS soil moisture data due to the use of GLDAS data for driving, which is reflected in both Figure 2 and Table 2, while the soil moisture provided by ERA5 itself is 0-7cm, which is somewhat different from the depth of the CLM simulation results, and the difference in driving data, so it may have resulted in a better agreement between the CLM simulation results and the ERA5 data to be less consistent than CLM with GLDAS data.

2) In figure 3, SIFyield, there are year variation. Do you check the relationship between SIFyield and PAR or other parameters?

Thank you for your question and my answer is as follows:

In order to better exclude the influence of meteorological factors such as temperature on the study, the following processing operations were performed during the data processing stage: (1) exclude pixels with a daily mean temperature <15°C (corresponding to the SIFyield, as well as pixels with SM and VPD were excluded), and (2) exclude pixels with a daily mean VPD <0.5 kPa. Through the above operations, we excluded as much as possible the influence of temperature and solar radiation on the results. In addition, we calculated SIFyield using PAR and fPAR to separate the overlapping effects of PAR and fPAR on ecosystem production. Although the above manipulations can try to avoid the effects of meteorological elements on the results, they cannot completely eliminate the effects of factors such as temperature, which is one of the limitations of this study.

Meanwhile, we have revised the Methods and Discussion sections, added the relevant descriptions in response to your questions, and added relevant literature to theoretically support the Methods section for better understanding.

3)    Unfortunately, for me, it is not clearly about your explanations about the value of binning in Figure 5 and others. Are Yearly, monthly and 8 days SM or VPD?   Difficult to understand the relationship between SM_bins and/or VPD_Bins and year, month and 8 days.

Re: Thank you for your question, and my answer is as follows:

Figure 5 shows the change of correlation between SM and VPD at different time scales, we calculated the correlation coefficients of the two elements at the annual scale, the correlation coefficients of the two elements at the monthly scale for the period 2001-2020... respectively, and the correlation coefficient of the two elements at the 8-day scale. It is only through such results that we can observe that the correlation between SM, VPD and SIFyield decreases as the time scales become more detailed. Since the minimum time scale of SIF data is 8-day scale, there may still be some correlation between SM and VPD at this scale. To separate the coupling of SM and VPD and discuss their effects on SIF individually, we utilized the method of data binning.

Specifically, we divided the SM and VPD at the 8-day scale into 10 bins based on their respective pixel values ranging from 10% to 100%. When using SM_bins as the y-value and VPD as the x-value, we found that the correlation coefficient between SM_bins and VPD is almost zero. The same result was obtained for the VPD split-box. Therefore, we reduced the correlation coefficient between SM and VPD by using the split-box calculation. This decoupled the coupling between VPD and SM. Based on your question, we have added explanations to the Methods and Results sections for better understanding.

4)    Can you also explain how to process binning? After figure 6 (line 252-271), can you explain the data binning can decouple the correlation between VPD and SIFyield? are figure 8 and figure 9 explained the decoupling? If so, I suggest you to add figure between VPD_SYFYield and SM.

Re: Thanks for your comments, we answer them:

(1)The data binning process is outlined as follows: For each pixel, we establish the 10th, 20th, and up to the 100th percentile thresholds for both SM and VPD. These thresholds are utilized to categorize the data into 10 distinct bins, corresponding to the percentile ranges of 0-10th, 10th-20th, through to 80-90th, and 90-100th for SM or VPD. This binning procedure maintains the temporal alignment of the datasets. Given that SM and VPD are substantially uncoupled within each specific SM or VPD bin, as shown in Figure 5, we are able to isolate the individual impacts of SM and VPD on the SIFyield. To ensure comparability across different spatial areas, the SIFyield time series for each pixel is normalized against the average SIFyield observed at the 90th percentile for that pixel.

(2)Data binning does not decouple VPD and SIFyield and SM and SIFyield correlation .The purpose of employing data binning in our analysis is to disentangle the relationship between soil moisture (SM) and vapor pressure deficit (VPD), acknowledging their evident correlation and respective impacts on the Solar-Induced Fluorescence yield (SIFyield). Recognizing that both SM and VPD significantly influence SIFyield, our data binning strategy aims to isolate the effects of SM and VPD. By categorizing data into bins based on predefined thresholds for SM and VPD, we effectively separate the influence of each variable. This decoupling facilitates a more focused examination of how each factor, independently, affects SIFyield, laying the groundwork for more detailed subsequent analyses.

5) In figure 9, the variation of VPD_Bins (from 70-80, 80-90, 90-100) be big. Do you explain why?

Re: Thank you for your question, my answer is as follows:

The correlation coefficients between soil moisture (SM) and solar-induced fluorescence yield (SIFyield) exhibit significant variability when the vapor pressure deficit (VPD) bins range from 70% to 100%. This fluctuation is primarily attributed to the categorization of 70%-100% VPD bins as representing an extreme state, characterized by elevated VPD levels. Such high VPD levels are indicative of extreme drought conditions, which likely lead to alterations in SIFyield. Consequently, these changes in SIFyield influence its correlation with SM. Furthermore, our findings corroborate the hypothesis that VPD exerts a more substantial impact on SIFyield than SM does in Southwest China, aligning with the primary conclusion of this study.

We have added the above analysis to the analysis of the results to facilitate the understanding of the results.

6)    In figure 10, is it only one point information. If so, you should add other area information.

Re: Thank you for your comments, I will answer them:

Figure 10 is not the only point. We divided the Southwest region into different vegetation types and averaged the calculations and statistics for each vegetation type region. So, Figure 10 is a result for the Southwest region as a whole.

7)    In line 323, you mention drought mechanisms, but there is no clear information about drought mechanisms and relationships with VPD and SM. And, explain drought mechanisms, it is also important to study the relationship between SM/VPD and temperature, PAR and rainfall as well as LULCC.

Re: Thank you for your question and my answer to your question is as follows:

(1)Soil moisture (SM) and vapor pressure deficit (VPD) serve as key indicators of drought, distinguishing between agricultural and meteorological droughts in areas of low SM and high VPD, respectively. These conditions significantly influence a region's ecosystem productivity. To understand the separate and combined effects of SM and VPD on this productivity, we employ data binning among other analytical techniques for our investigation.

(2)SM and VPD are closely related to temperature, PAR, and rainfall, which also affect ecosystem productivity, so we performed some data screening to eliminate the effects of temperature, PAR, and other factors as much as possible. We have added the detailed steps in the Methods section. In addition to this, data binning not only decouples the correlation between SM and VPD, but also does not affect their matching time, according to Liu et al.'s article, the meteorological conditions of SM and VPD are kept the same under the corresponding binning, which also helps us to better control the variables while discussing the effect of the two separately on SIFyeild.

(3)Land use change significantly influences the outcomes of this study. In Figure 10, we examine how variations in soil moisture (SM) and vapor pressure deficit (VPD) impact the yield of solar-induced fluorescence (SIF) across different vegetation types. Our analysis reveals notable differences in how the productivity of these vegetation types is affected by changes in SM and VPD.

8) In line 341, you should spell out CLM4.5 and explain why you adopt CLM4.5.

Re: Thank you very much for your suggestion, I have added some of what you suggested to the methods section.

“In this study, the study area's soil moisture and other variables were simulated using the land surface process model CLM4.5 from 2005 to 2020. The CLM4.5 model is an internationally recognized and highly advanced model for studying land surface processes. It is the land surface module of the Common Earth System Model [29, 30]. The CLM4.5 model realizes the heterogeneity of land surface space by nesting grid, that is, the grid contains a variety of land individuals, snow, soil columns and vegetation functional types [31]. The CLM4.5 model stands out due to its nested grid capability, which enables the simulation of intricate surface features. Moreover, it offers a de-tailed and complex depiction of processes such as soil moisture dynamics, plant growth and mortality, and the exchange of water and energy between vegetation and the atmosphere [32]. CLM4.5 encompasses extensive simulations of phenomena like root water uptake, soil evaporation, plant transpiration, and permafrost processes [33]. As a result, it potentially delivers more accurate simulations of soil moisture dynamics under specific conditions compared to other models. In this study, we use the CLM4.5 model to simulate soil moisture and evapotranspiration data in southwest China.”

Round 2

Reviewer 1 Report

Comments and Suggestions for Authors

N/A

Author Response

Thank you very much for your review comments, they are a great enhancement to the article.

Reviewer 2 Report

Comments and Suggestions for Authors

The authors have taken my observations into account and the work is much improved. I congratulate the authors for their work.

Author Response

(The authors gave the same response as above.)

Reviewer 3 Report

Comments and Suggestions for Authors

Better to add some explanation about the definition based on author's reply as follow.

(1)The data binning process is outlined as follows: For each pixel, we establish the 10th, 20th, and up to the 100th percentile thresholds for both SM and VPD. These thresholds are utilized to categorize the data into 10 distinct bins, corresponding to the percentile ranges of 0-10th, 10th-20th, through to 80-90th, and 90-100th for SM or VPD. This binning procedure maintains the temporal alignment of the datasets. Given that SM and VPD are substantially uncoupled within each specific SM or VPD bin, as shown in Figure 5, we are able to isolate the individual impacts of SM and VPD on the SIFyield. To ensure comparability across different spatial areas, the SIFyield time series for each pixel is normalized against the average SIFyield observed at the 90th percentile for that pixel.

Author Response

Better to add some explanation about the definition based on author's reply as follow.

(1)The data binning process is outlined as follows: For each pixel, we establish the 10th, 20th, and up to the 100th percentile thresholds for both SM and VPD. These thresholds are utilized to categorize the data into 10 distinct bins, corresponding to the percentile ranges of 0-10th, 10th-20th, through to 80-90th, and 90-100th for SM or VPD. This binning procedure maintains the temporal alignment of the datasets. Given that SM and VPD are substantially uncoupled within each specific SM or VPD bin, as shown in Figure 5, we are able to isolate the individual impacts of SM and VPD on the SIFyield. To ensure comparability across different spatial areas, the SIFyield time series for each pixel is normalized against the average SIFyield observed at the 90th percentile for that pixel

Re: Thank you very much for your comments, we have refined the definitions in the data binning section.

“SM and VPD are strongly coupled, and the experiment require a reduction in the coupling of the two elements to better investigate the influence of each on regional ecosystem production [10]. This experiment employs a data-binning approach [36, 37] to reduce the strong coupling correlation between SM and VPD, which is performed as follows: The data binning process is outlined as follows: For each pixel, we establish the 10th, 20th, and up to the 100th percentile thresholds for both SM and VPD. These thresholds are utilized to categorize the data into 10 distinct bins, corresponding to the percentile ranges of 0-10th, 10th-20th, through to 80-90th, and 90-100th for SM or VPD. This binning procedure maintains the temporal alignment of the datasets. Given that SM and VPD are substantially uncoupled within each specific SM or VPD bin, as shown in Figure 5, we are able to isolate the individual impacts of SM and VPD on the SIFyield. To ensure comparability across different spatial areas, the SIFyield time series for each pixel is normalized against the average SIFyield observed at the 90th percen-tile for that pixel.”
